# Association of per- and polyfluoroalkyl substances with constipation: The National Health and Nutrition Examination Survey (2005–2010)

Yifan Zhao[1,2,3☯], Ke Pu[4☯], Ya Zheng[2,3], Yuping Wang[2,3], Jun Wang[2,3]*, Yongning Zhou[2,3]*

1 The First Clinical Medical College, Lanzhou University, Lanzhou, China, 2 Department of Gastroenterology, The First Hospital of Lanzhou University, Lanzhou, China, 3 Gansu Province Clinical Research Center for Digestive Diseases, The First Hospital of Lanzhou University, Lanzhou, China, 4 Department of Gastroenterology, Affiliated Hospital of North Sichuan Medical College, Sichuan, China

☯ These authors contributed equally to this work.
* zhouyn@lzu.edu.cn (YZ); ldyy_wangjun@lzu.edu.cn (JW)

**Data Availability Statement:** All data files are available from the NHANES database (website: https://www.cdc.gov/nchs/nhanes/index.htm).

## Abstract

### Background

The impact of per- and polyfluoroalkyl substances (PFAS) on constipation, as mediated through gastrointestinal absorption and perturbations to the intestinal microecology, remains poorly understood.

### Objective

This study seeks to explain the relationship between PFAS and constipation.

### Methods

A total of 2945 adults from the National Health and Nutrition Examination Survey (NHANES) 2005–2010 were included in this study. Constipation was defined using the Bristol Stool Form Scale (BSFS) based on stool consistency. The relationship between PFAS and constipation was evaluated using weighted logistic regression and restricted cubic spline (RCS) analysis, while adjusting for confounding variables.

### Results

The weighted median concentration of total PFAS (ΣPFAS) was significantly lower in individuals with constipation (19.01 μg/L) compared to those without constipation (23.30 μg/L) (p < 0.0001). Subgroup analysis revealed that the cumulative effect of PFAS was more pronounced in the elderly, men, individuals with obesity, high school education or equivalent, and high-income individuals (p < 0.05). After adjusting for confounding factors, multivariable analysis demonstrated an inverse association between PFOA [OR (95% CI), 0.666 (0.486,0.914)] and PFHxS [OR (95% CI), 0.699(0.482,1.015)], and constipation. None of the personal and lifestyle factors showed a significant correlation with this negative

**Funding:** Yongning Zhou received funding from the National Natural Science Foundation of China (No. 71964021), and Ya Zheng received funding from the Natural Science Foundation of Gansu Province (NO. 23JRRA0939). The funders had no role in study design, data collection and analysis, decision to publish, or preparation of the manuscript.

**Competing interests:** The authors have declared that no competing interests exist.

**Abbreviations:** PFAS, Per- and polyfluoroalkyl substances; PFOA, Perfluorooctanoate; PFOS, Perfluorooctane sulfonate; PFHxS, Perfluorohexane sulfonate; PFDeA, Perfluorodecanoate; PFNA, Perfluorononanoate; MPAH, 2- (N-methyl-perfluorooctane sulfonamido) acetate (Me-PFOSA-AcOH); ΣPFAS, The total concentrations of PFOA, PFOS, PFHxS, PFDeA, PFNA, MPAH; NHANES, National Health and Nutrition Examination Survey; NCHS, National Center for Health Statistics; CDC, Centers for Disease Control and Prevention; GED, General Educational Development; POPs, Persistent Organic Pollutants; BMI, Body Mass Index; PIR, Poverty-to-income ratio; HEI-2015, Healthy Eating Index-2015; MET, Metabolic Equivalent; IBD, Inflammatory bowel disease; BSFS, Bristol Stool Form Scale; LOD, limit of detection; IPQA, International Physical Activity Questionnaire; IQR, Interquartile Range; RCS, Restricted Cubic Spline; OR, Odd Ratio; CI, Confidence Interval.

association, as confirmed by subgroup analysis and interaction testing (p for interaction > 0.05). The RCS analysis demonstrated a linear inverse relationship between PFAS levels and constipation.

## Conclusion

The findings of this study provide evidence of a significant inverse correlation between serum concentrations of PFAS, particularly PFOA and PFHxS, and constipation.

## 1. Introduction

Per- and polyfluoroalkyl substances (PFAS) are a group of man-made chemicals that have gained significant attention in recent years due to their persistence, bioaccumulation, and potential health impacts [1]. PFAS encompass a large group of chemicals, including perfluorooctanoic acid (PFOA) and perfluorooctanesulfonic acid (PFOS), which are among the most extensively studied compounds. However, numerous other PFAS variants exist, and newer replacement compounds are also being used. PFAS are widely used in various industrial and consumer products such as firefighting foam, non-stick cookware, water-repellent clothing, and stain-resistant carpets due to their unique properties, including oil and water repellency [2]. However, due to their widespread use and persistence, PFAS can migrate in the environment through various pathways, including air deposition, wastewater discharges, industrial releases, and the use of PFAS-containing products [3]. Moreover, PFAS can interact with soil and sediments, bind to organic matter, and persist for a long time [4]. Once released, they can travel long distances, contaminate water sources, and enter the food chain. PFAS compounds have been detected in various environmental compartments such as surface waters, groundwater, soil, air, and biota. They have also been found in human blood, urine, breast milk, and other tissues [5]. SO it has recently been recognized as a class of persistent organic pollutants (POPs) [6].

Recent research has highlighted the adverse effects of PFAS on various health issues, including immunological, metabolic, cardiovascular, liver, renal problems, and cancer [7]. Furthermore, studies have indicated that PFAS can disrupt the intestinal barrier, alter microbial ecology, and contribute to metabolic dysfunction, leading to digestive illnesses [8]. Specifically, a significant body of research has explored the association between PFAS exposure and inflammatory bowel disease (IBD), with a particular emphasis on ulcerative colitis [9–14]. Moreover, it has been suggested that PFAS could induce intestinal environmental disturbances, which may play a role in the development of constipation.

Constipation is a prevalent digestive symptom that negatively impacts quality of life, with a pooled prevalence of 10.1% (8.7–11.6; I2 = 98.2%) reported in recent meta-analysis [15]. Approximately 11–20% of the adult population experience constipation annually [16] and a significant proportion of these individuals seek medical care but are often dissatisfied with the available treatment options [17]. Considering that environmental and lifestyle factors can influence constipation [18], it is crucial to investigate the correlation between PFAS and constipation symptoms. However, a comprehensive understanding of how PFAS contributes to constipation is currently lacking. Our objective was to analyze the relationship between PFAS and constipation in order to understand the extent to which PFAS contributes to the development of this condition.

## 2. Method

### 2.1 Study population

Data were extracted on March 6, 2023 from the NHANES (National Health and Nutrition Examination Survey) database. NHANES is a comprehensive and representative population health survey database administered by the National Center for Health Statistics (NCHS), which is part of the Centers for Disease Control and Prevention (CDC). It collects data on various aspects of health, lifestyle, nutritional status, environmental exposure, and more, providing valuable information for public health research, policy formulation, program design, and expanding national health knowledge [19]. In this study, we analyzed data from 31,034 samples collected over three NHANES cycles (spanning from January of the previous year to December of the corresponding year): 2005–2006, 2007–2008, and 2009–2010. The dataset included information on intestinal symptoms, as well as PFAS test results. A total of 27,993 samples were excluded from the study due to incomplete data, and an additional 96 samples were excluded due to colon cancer or pregnancy. Ultimately, 2,945 samples were included in the study (Fig 1).

NHANES investigation protocol was approved by the Institutional Research Ethics Review Committee of the CDC's National Center for Health Statistics. All participants provided written informed consent, the study by the NCHS research ethics review committee (https://wwwn.cdc.gov/nchs/nhanes/default.aspx) for approval.

### 2.2 Definition of constipation

Constipation was defined based on stool consistency using the NHANES database [20]. The Bristol Stool Form Scale (BSFS) was employed to provide an objective description of stool texture and estimate stool consistency. The scale includes seven grades, ranging from 1 to 7, each corresponding to a distinct stool morphology. (Type 1: individual hard lumps, such as nuts; type 2: sausage-like, but lumpy; type 3: like a sausage, but with cracks on the surface; type 4: like a sausage or snake, smooth and soft; type 5: soft spots with clear edges; type 6: fluffy fragments with rough edges, fecal mush; and type 7: aqueous samples, no solid debris). Constipation was defined as having a BSFS score of 1 or 2, indicating hard or lumpy stools with a dry surface [15].

### 2.3 Plasma PFAS assessment

The estimation of PFAS concentrations in the study utilized the on-line solid-phase extraction-high performance liquid chromatography-isotope dilution-tandem mass spectrometry techniques at CDC laboratories. Detailed analytical procedures can be found on the NHANES website: https://www.cdc.gov/nchs/nhanes/. Concentrations below the limit of detection (LOD) were imputed as $LOD/\sqrt{2}$, following the method employed in CDC's National Report on Human Exposure to Environmental Chemicals [21]. Among the 12 PFAS assessed across three cycles, only six with a detection rate greater than or equal to 70% were included in this study: 2-(N-methyl-perfluorooctane sulfonamido) acetate (Me-PFOSA-AcOH), perfluorohexane sulfonate (PFHxS), perfluorooctane sulfonate (PFOS), perfluorooctanoate (PFOA), perfluorononanoate (PFNA), and perfluorodecanoate (PFDeA). Additionally, the total concentrations of these six PFAS were calculated and represented as ∑PFAS.

### 2.4 Covariates

The study also incorporated the following factors: age (categorized as 20–39, 40–64, and ≥65 years), gender (male and female), race (Other Hispanic and other race, Mexican American,

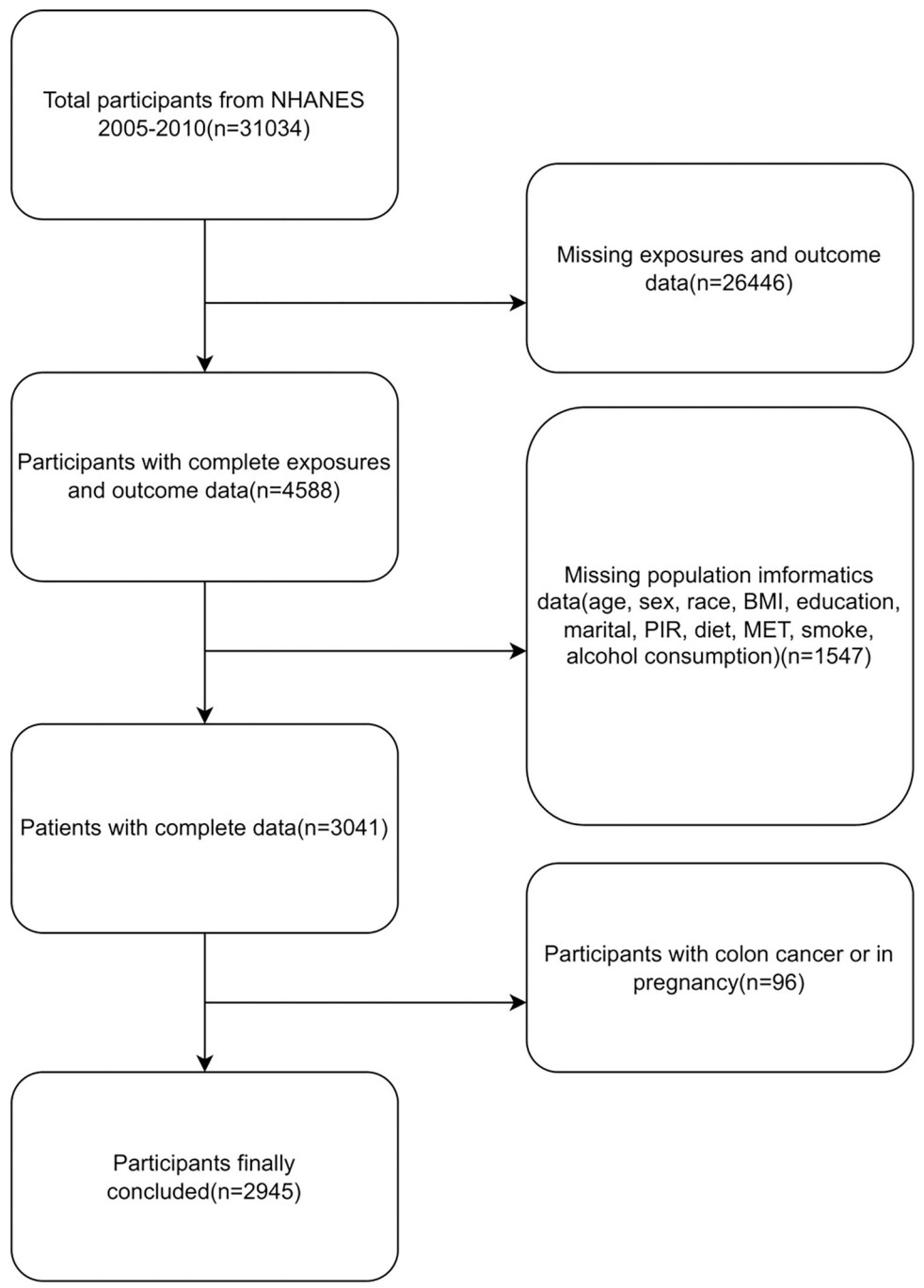

**Fig 1. Participant flowchart for the examination of association between PFAS exposure and constipation in adult participants of the NHANES Survey (2005–2010).**

non-Hispanic Black, non-Hispanic White), body mass index (BMI) calculated as weight (kg) divided by height squared (m^2) with categories including under or normal range (<25 kg/m^2) and overweight (≥25 kg/m^2), educational level (categories: less than high school—completed less than 9th grade or 9-11th grade; high school or equivalent—high school graduate/GED or equivalent; above high school—some college or associate's degree and college

graduate or above), marital status (partner status: married, living with partner; solitary status: widowed, divorced, separated, and never married), and the Poverty-to-Income Ratio (PIR), which serves as a measure of socioeconomic status (low income: PIR ≤ 1.30; middle income: PIR 1.31–3.50; high income: PIR > 3.50).

Diet quality is assessed using the HEI-2015 (Healthy Eating Index-2015) rating scale, which has a maximum score of 100 points. Higher scores on the HEI-2015 indicate better overall meal quality. The scoring is categorized as follows: a total score of 90 to 100 falls into group A, 80 to 89 is in group B, 70 to 79 is in group C, 60 to 69 is in group D, and a score of 0 to 59 is in group F [22]. Due to the limited number of samples in groups A, B, and C, they are combined into a single group for analysis. After merging, dietary quality was categorized into three groups: the ABC group (70–100), group D (60–69), and group F (0–59). Total physical activity was measured using The International Physical Activity Questionnaire (IPQA) [23]. Activity levels were categorized as follows: low activity, defined as < 600 MET-min/week; moderate activity, defined as 600–3000 MET-min/week; and high activity, defined as > 3000 MET-min/week [24, 25]. Smoking status was categorized as follows: never-smoked, indicating individuals who have smoked fewer than 100 cigarettes in their lifetime; previous smokers, representing those who have smoked more than 100 cigarettes in their lifetime but no longer smoke; and current smokers, referring to individuals who have smoked more than 100 cigarettes and currently smoke every day or on several days [26]. Drinking status was classified as follows: Never, for individuals who have consumed fewer than 12 alcoholic drinks in their lifetime; Former, for those who have consumed at least 12 alcoholic drinks in their lifetime but have consumed fewer than 12 alcoholic drinks in the previous year; and Now, for individuals who have consumed more than 12 alcoholic drinks in the previous year [27].

## 2.5 Statistical analysis

Data management and analysis were conducted using R software (version 4.2.3). Normally distributed continuous variables are reported as mean and standard deviation (SD), non-normally distributed continuous variables are reported as median (IQR = Q25 (first quartile), IQR = Q75 (third quartile)), and categorical variables are presented as number of cases (n) and frequency (%). Group comparisons for normally distributed variables were assessed using Student's t-test, while for skewed variables, the Wilcoxon rank-sum test was employed. The chi-square test was used to analyze rates for categorical variables.

We conducted subgroup analyses to explore potential associations, stratifying by age, sex, race, BMI, marital status, education level, income, activity, diet quality, smoking status, and alcohol consumption. Additionally, we assessed the correlations between PFAS and the prevalence of constipation in these subgroups using multivariable logistic regression.

PFAS concentrations were analyzed after applying a natural log transformation. Each of the six PFAS compounds was categorized into three concentration groups (Q1-Q3). We examined the relationship between exposure variables and constipation using logistic regression. Three models were constructed: the unadjusted model and two adjusted models to account for potential covariates. Model A included adjustments for age, gender, race, body mass index, educational level, and marital status. Model B, built upon Model A, also included adjustments for diet pattern, activity level, smoking status, and alcohol consumption. Additionally, we conducted tests for trends by creating a new continuous variable based on the three concentration groups (Q1-Q3) to explore any linear relationship between PFAS concentrations and constipation. The results were reported as odds ratios (ORs) and 95% confidence intervals (CIs).

Since the correlation between PFHxS and PFOA has been found more prominent, correlations between these two substances and the prevalence of constipation in subgroups were

evaluated using multivariable logistic regression. Additionally, to obtain a comprehensive understanding of the relationship between the variables and the outcome, we performed a restricted cubic spline (RCS) analysis. The dataset was adjusted for potential confounding factors, including age, gender, race, body mass index, educational level, marital status, PIR, diet pattern, activity, smoke status, and alcohol consumption.

## 3. Results

### 3.1 Characteristics of the study participants

A total of 2945 participants from the NHANES database between 2005 and 2010 were included in this study (Table 1).

Among the study population, 203 individuals were identified as constipation patients. No significant differences were observed between patients and non-patients regarding age, ethnicity, physical activity, diet pattern, smoking status, alcohol consumption, and year cycle (p > 0.05). However, compared to non-constipated individuals, constipation patients exhibited a higher proportion of women, lower body mass index, and lower income level (p < 0.05). Regarding the distribution characteristics of PFAS, it was found that most PFAS (including PFOA, PFOS, PFHxS, PFDeA, and PFNA) demonstrated a significant reduction in constipation patients (p < 0.05). Additionally, the total PFAS (ΣPFAS) level was significantly lower in patients with constipation (19.01 μg/L) compared to the non-constipated group (23.30 μg/L) (p < 0.0001).

### 3.2 Distribution characteristics of PFAS

Table 2 presents the levels of PFAS in different groups of people. Overall, there was a noticeable increase in PFAS concentration with age, particularly for PFHxS, PFOS, and MPAH (p < 0.001). Males exhibited higher levels of most PFAS compared to females (p < 0.001). Moreover, the levels of PFAS varied significantly among different race groups, income levels, activity categories, as well as among subgroups based on smoking and alcohol consumption habits.

### 3.3 Analysis of the association between PFAS and the risk of constipation

The data in Table 3 reveals a negative correlation between PFAS and constipation. In comparison to the lowest quartile (Q1), individuals in the highest quartiles (Q3) of PFOA [OR (95% CI), 0.562(0.421,0.751), p <0.001], PFHxS [OR (95% CI), 0.585(0.428,0.800), p = 0.001], and PFOS [OR (95% CI), 0.650(0.483,0.875), p = 0.006] exhibited a reduced risk of constipation in the unadjusted model. However, the association between PFOS and constipation disappeared when adjusting for multiple potential influencing factors (model A and model B). In contrast, the correlation between PFOA [model A in Q3: OR (95% CI), 0.666(0.472,0.939), p = 0.022; model B in Q3: OR (95% CI), 0.666(0.486,0.914), p = 0.013] and PFHxS [model A in Q2: OR (95% CI), 0.569(0.376,0.863), p = 0.011; model B in Q2: OR (95% CI), 0.578(0.412,0.811), p = 0.002] remained significant. The trend analysis indicated a decreasing trend in the association between PFOA and constipation (P in unadjusted model: <0.001, P in model A: 0.023, P in model B: 0.01), while an increasing trend was observed in the case of PFHxS and constipation (P in unadjusted model: <0.001, P in model A: 0.053, P in model B: 0.049).

### 3.4 Analysis of the association between PFAS and the risk of constipation in subgroups

The associations between PFAS (continuous log-transformed PFOA and PFHxS) and constipation in various subgroups are depicted in Figs 2 and 3.

**Table 1. Weighted characteristics of participants with constipation and without constipation in the NHANES database, NHANES 2005–2010.**

| Variable | Non-Constipation(n = 2742) | Constipation(n = 203) | p value |
|---|---|---|---|
| Gender, N (%) | | | **<0.001** |
| Male | 1281(47.21) | 121(64.52) | |
| Female | 1461(52.79) | 82(35.48) | |
| Age (yr), mean (SD) | 45.24(0.50) | 43.27(1.40) | 0.25 |
| Age (yr), N (%) | | | 0.77 |
| 20-39(Young) | 984(39.87) | 85(42.73) | |
| 40-64(Middle aged) | 1204(46.94) | 84(46.28) | |
| > = 65(Elderly) | 554(13.20) | 34(10.99) | |
| Ethnicity, N (%) | | | 0.32 |
| Other Hispanic and Other Race | 323(8.72) | 24(8.76) | |
| Mexican American | 450(6.89) | 37(8.22) | |
| Non-Hispanic Black | 470(8.68) | 43(11.99) | |
| Non-Hispanic White | 1499(75.71) | 99(71.02) | |
| Education, N (%) | | | **0.05** |
| Less than high-school | 621(14.68) | 55(18.33) | |
| High-school or equivalent | 584(21.14) | 57(26.94) | |
| Above high-school | 1537(64.18) | 91(54.74) | |
| Marital status, N (%) | | | **0.03** |
| Partner status | 1723(65.96) | 113(57.79) | |
| Solitary status | 1019(34.04) | 90(42.21) | |
| PIR, N (%) | | | **0.02** |
| <1.30 | 700(16.61) | 72(23.75) | |
| 1.30–3.50 | 1027(34.24) | 81(40.30) | |
| >3.50 | 1015(49.15) | 50(35.95) | |
| BMI((kg/m2), N (%) | | | **0.004** |
| <25(Lean/Normal) | 847(33.01) | 80(46.20) | |
| ≥25(Over weight/obesity) | 1895(66.99) | 123(53.80) | |
| MET(MET-min/week), N (%) | | | 0.95 |
| < 600 | 746(27.49) | 55(26.58) | |
| 600–3000 | 1132(42.48) | 83(43.71) | |
| >3000 | 864(30.03) | 65(29.70) | |
| HEI-2015(points), N (%) | | | 0.63 |
| 70-100(A/B/C level) | 261(9.12) | 20(11.16) | |
| 60-69(D level) | 479(17.30) | 34(14.95) | |
| 0-59(F level) | 2002(73.58) | 149(73.89) | |
| Smoke status, N (%) | | | 0.25 |
| Never | 1446(53.56) | 120(60.18) | |
| Former | 705(24.85) | 45(20.52) | |
| Now | 591(21.58) | 38(19.30) | |
| Drinking status, N (%) | | | 0.19 |
| Never | 314(9.38) | 36(13.84) | |
| Former | 507(15.28) | 32(13.99) | |
| Now | 1921(75.34) | 135(72.17) | |
| Year cycle, N (%) | | | 0.4 |
| 2005–2006 | 865(36.53) | 54(31.50) | |
| 2007–2008 | 913(31.44) | 76(38.10) | |
| 2009–2010 | 964(32.02) | 73(30.40) | |

*(Continued)*

**Table 1.** (Continued)

| Variable | Non-Constipation(n = 2742) | Constipation(n = 203) | p value |
|---|---|---|---|
| PFAS, median (IQR) | | | |
| PFOA | 4.10(2.70,5.80) | 3.40(2.20,5.00) | < **0.001** |
| PFOS | 14.00(8.70,22.20) | 11.40(6.80,18.50) | < **0.001** |
| PFHxS | 1.90(1.10,3.20) | 1.30(0.90,2.70) | < **0.001** |
| PFDeA | 0.30(0.20,0.50) | 0.30(0.20,0.40) | **0.01** |
| PFNA | 1.20(0.82,1.72) | 1.07(0.82,1.56) | **0.02** |
| MPAH | 0.30(0.12,0.50) | 0.30(0.12,0.60) | 0.71 |
| ΣPFAS | 23.30(15.26,33.84) | 19.01(12.36,30.20) | < **0.001** |

N represents unweighted counts.

Percentages are weighted to the American population.

Categorical variables were expressed as N (%). Normally distributed continuous variables are presented as the mean (SD), non-normally distributed continuous variables are presented as median (IQR)

While there were significant differences in PFAS distribution among different populations, interaction tests indicated that the relationship between PFAS and constipation was not statistically different across subgroups. This suggests that factors such as age, sex, race, BMI, education, income level, marital status, diet, physical activity, smoking, and drinking status did not significantly affect the observed positive correlation (p for interaction > 0.05).

### 3.5 The association between individual chemicals and constipation displayed by RCS analysis

By utilizing restricted cubic spline (RCS) analysis (Fig 4A and 4B), we further explored the linear and nonlinear relationships between ln (PFAS) and constipation, while adjusting for age, gender, race, body mass index, educational level, marital status, PIR, diet pattern, activity, smoking status, and alcohol consumption. In the association between constipation and ln (PFOA), we observed a linear relationship, with the prevalence of constipation increasing as ln (PFOA) values increased (P-non-linear = 0.128, P-overall = 0.054). Likewise, for ln (PFHxS), a linear relationship was also noted (P-non-linear = 0.493, P-overall = 0.217). The negative correlation of constipation risk diminishes at higher levels of these compounds.

## 4. Discussion

In our cross-sectional study, we observed a negative association between higher PFAS levels, specifically PFOA and PFHxS, and constipation. Subgroup analysis and interaction tests indicated that this association remained consistent across various subgroups, despite significant differences in PFAS distribution among different populations.

Consistent with previous studies, our findings support that the prevalence of constipation tends to be higher among women [28], individuals with normal or lower body mass index [29], and those with middle to low income levels [30].

Regarding the distribution characteristics of PFAS, our study corroborates previous findings that PFAS accumulation tends to be higher in older individuals, which can be attributed to the longer half-life of PFAS [31], leading to their gradual build-up over time. Furthermore, a significant decrease in PFAS concentration was observed in females [32] potentially explained by factors such as menstrual blood loss or the influence of female hormones on the expression of organic anion transporters, which are involved in the kidney clearance and metabolism of

**Table 2. Distribution of PFAS.**

| Subgroups | PFOA | PFOS | PFHxS | PFDeA | PFNA | MPAH | ΣPFAS |
|---|---|---|---|---|---|---|---|
| Gender | | | | | | | |
| Female | 4.60 (3.37, 6.58) | 17.00 (11.40, 25.10) | 2.30 (1.50, 3.80) | 0.30 (0.20, 0.50) | 1.30 (0.90, 1.89) | 0.30 (0.12, 0.54) | 27.13 (19.39, 38.14) |
| Male | 3.30 (2.20, 5.00) | 11.04 (6.60, 17.55) | 1.30 (0.80, 2.30) | 0.30 (0.15, 0.40) | 1.07 (0.74, 1.60) | 0.30 (0.12, 0.50) | 18.21 (12.03, 27.77) |
| p value | **<0.001** | **<0.001** | **<0.001** | **<0.001** | **<0.001** | **0.005** | **<0.001** |
| Age (yr) | | | | | | | |
| 20-39(Young) | 3.90 (2.50, 5.70) | 12.19 (7.30, 18.70) | 1.70 (0.96, 3.10) | 0.30 (0.20, 0.50) | 1.15 (0.80, 1.64) | 0.30 (0.12, 0.50) | 21.1(13.6,30.2) |
| 40-64(Middle aged) | 4.10 (2.70, 5.80) | 14.40 (9.10, 22.60) | 1.80 (1.10, 3.10) | 0.30 (0.20, 0.50) | 1.23 (0.90, 1.80) | 0.30 (0.12, 0.50) | 23.3(15.5,35.5) |
| > = 65(Elderly) | 4.20 (2.90, 6.10) | 18.50 (11.94, 29.59) | 2.10 (1.30, 3.40) | 0.30 (0.20, 0.50) | 1.23 (0.82, 1.80) | 0.40 (0.20, 0.90) | 28.5(19.5,42.8) |
| p value | 0.066 | **<0.001** | **<0.001** | 0.7 | **0.033** | **<0.001** | **<0.001** |
| Ethnicity | | | | | | | |
| Other Hispanic and Other Race | 3.48 (2.30, 5.22) | 11.66 (6.90, 19.80) | 1.40 (0.80, 2.43) | 0.30 (0.20, 0.60) | 1.15 (0.82, 1.89) | 0.30 (0.12, 0.40) | 19.24(12.32,31.30) |
| Mexican American | 3.30 (2.00, 4.61) | 9.99 (6.36, 16.80) | 1.50 (0.80, 2.80) | 0.30 (0.14, 0.40) | 1.00 (0.70, 1.48) | 0.20 (0.12, 0.30) | 16.98(11.18,25.77) |
| Non-Hispanic Black | 3.70 (2.40, 5.40) | 16.60 (9.20, 25.58) | 1.80 (0.99, 3.31) | 0.40 (0.20, 0.60) | 1.39 (0.90, 1.89) | 0.30 (0.12, 0.50) | 25.91(16.05,37.74) |
| Non-Hispanic White | 4.20 (2.80, 5.90) | 14.30 (9.09, 22.18) | 1.90 (1.10, 3.20) | 0.30 (0.20, 0.50) | 1.20 (0.82, 1.72) | 0.30 (0.20, 0.50) | 23.60(15.69,34.39) |
| p value | **<0.001** | **<0.001** | **<0.001** | **<0.001** | **<0.001** | **<0.001** | **<0.001** |
| Education | | | | | | | |
| Less than high-school | 3.50 (2.30, 5.50) | 12.50 (7.30, 19.69) | 1.70 (1.00, 2.90) | 0.30 (0.20, 0.50) | 1.20 (0.82, 1.80) | 0.30 (0.12, 0.50) | 20.23(13.06,31.07) |
| High-school or equivalent | 4.30 (2.80, 6.00) | 15.60 (9.80, 24.13) | 2.00 (1.20, 3.40) | 0.30 (0.20, 0.50) | 1.23 (0.82, 1.80) | 0.30 (0.12, 0.60) | 25.25(16.85,36.60) |
| Above high-school | 4.10 (2.70, 5.70) | 13.60 (8.60, 21.80) | 1.80 (1.10, 3.10) | 0.30 (0.20, 0.50) | 1.20 (0.82, 1.72) | 0.30 (0.12, 0.50) | 22.70(15.18,33.11) |
| p value | **0.001** | **<0.001** | **0.007** | 0.5 | 0.4 | 0.4 | **<0.001** |
| Marital status | | | | | | | |
| Partner status | 4.00 (2.60, 5.80) | 14.26 (8.60, 22.41) | 1.80 (1.00, 3.20) | 0.30 (0.20, 0.50) | 1.23 (0.82, 1.80) | 0.30 (0.12, 0.50) | 23.36(14.92,34.53) |
| Solitary status | 4.00 (2.80, 5.70) | 13.30 (8.40, 21.43) | 1.80 (1.10, 3.10) | 0.30 (0.20, 0.44) | 1.15 (0.82, 1.69) | 0.30 (0.20, 0.60) | 22.24(15.35,32.94) |
| p value | 0.4 | 0.074 | 0.8 | 0.2 | 0.3 | **0.04** | 0.3 |
| PIR | | | | | | | |
| <1.30 | 3.50 (2.27, 5.10) | 11.60 (7.20, 19.65) | 1.60 (0.90, 2.90) | 0.30 (0.20, 0.40) | 1.07 (0.74, 1.56) | 0.30 (0.12, 0.50) | 19.86(13.00,30.27) |
| 1.30–3.50 | 3.90 (2.60, 5.70) | 13.80 (8.30, 22.60) | 1.80 (1.00, 3.20) | 0.30 (0.20, 0.50) | 1.15 (0.82, 1.72) | 0.30 (0.12, 0.50) | 23.01(14.57,34.66) |
| >3.50 | 4.30 (2.90, 5.90) | 14.56 (9.80, 22.20) | 1.90 (1.20, 3.20) | 0.30 (0.20, 0.50) | 1.23 (0.90, 1.80) | 0.30 (0.12, 0.50) | 24.29(16.50,34.30) |
| p value | **<0.001** | **<0.001** | **0.003** | **0.006** | **<0.001** | 0.3 | **<0.001** |
| BMI((kg/m2) | | | | | | | |
| <25(Lean/Normal) | 4.00(2.60,5.50) | 13.25 (8.60, 20.90) | 1.70 (1.00, 3.10) | 0.30 (0.20, 0.50) | 1.15 (0.82, 1.64) | 0.30 (0.12, 0.50) | 21.90(14.90,32.39) |
| ≥25(Over weight/obesity) | 4.10(2.70,5.90) | 14.30 (8.50, 22.65) | 1.90 (1.10, 3.20) | 0.30 (0.20, 0.50) | 1.23 (0.82, 1.80) | 0.30 (0.12, 0.50) | 23.60(15.12,34.53) |
| p value | 0.079 | 0.1 | 0.2 | 0.11 | 0.064 | 0.4 | 0.09 |
| MET(MET-min/week) | | | | | | | |
| < 600 | 4.20 (2.70, 6.33) | 16.40 (9.80, 25.30) | 1.80 (1.00, 3.20) | 0.30 (0.20, 0.50) | 1.15 (0.80, 1.77) | 0.30 (0.20, 0.60) | 25.10(16.16,38.20) |
| 600–3000 | 3.90 (2.60, 5.60) | 13.10 (8.10, 20.70) | 1.70 (1.00, 3.00) | 0.30 (0.20, 0.50) | 1.15 (0.80, 1.64) | 0.30 (0.12, 0.50) | 22.04(14.40,31.91) |
| >3000 | 4.10 (2.70, 5.80) | 13.41 (8.60, 20.83) | 2.00 (1.20, 3.40) | 0.30 (0.20, 0.50) | 1.23 (0.90, 1.80) | 0.30 (0.12, 0.40) | 23.07(15.32,32.44) |
| p value | 0.085 | **<0.001** | **0.004** | **0.014** | **<0.001** | **<0.001** | **0.002** |
| HEI-2015(points) | | | | | | | |
| 70-100(A/B/C level) | 3.70 (2.50, 5.40) | 12.48 (7.80, 20.33) | 1.90 (1.20, 3.30) | 0.30 (0.20, 0.40) | 1.10 (0.74, 1.56) | 0.30 (0.12, 0.50) | 21.77(14.20,30.40) |
| 60-69(D level) | 3.80 (2.50, 5.60) | 13.70 (8.60, 21.40) | 1.70 (1.00, 3.18) | 0.30 (0.20, 0.50) | 1.15 (0.82, 1.72) | 0.30 (0.12, 0.50) | 22.88(14.73,33.11) |
| 0-59(F level) | 4.10 (2.70, 5.90) | 14.00 (8.70, 22.30) | 1.90 (1.10, 3.10) | 0.30 (0.20, 0.50) | 1.20 (0.82, 1.80) | 0.30 (0.12, 0.50) | 23.17(15.23,34.30) |
| p value | 0.065 | 0.2 | 0.5 | 0.13 | 0.11 | 0.3 | 0.2 |
| Smoke status | | | | | | | |
| Never | 3.97 (2.51, 5.60) | 13.75 (8.30, 21.40) | 1.70 (1.00, 3.00) | 0.30 (0.20, 0.50) | 1.20 (0.82, 1.72) | 0.30 (0.12, 0.50) | 20.40 (13.10, 30.10) |
| Former | 4.10 (2.80, 6.00) | 14.70 (8.97, 23.50) | 1.90 (1.10, 3.20) | 0.30 (0.20, 0.50) | 1.23 (0.90, 1.80) | 0.30 (0.20, 0.60) | 22.24 (14.17, 32.40) |
| Now | 4.13 (2.80, 6.20) | 13.10 (8.40, 21.72) | 2.00 (1.20, 3.50) | 0.30 (0.20, 0.50) | 1.20 (0.82, 1.72) | 0.30 (0.20, 0.60) | 20.50 (13.40, 31.09) |

(*Continued*)

**Table 2.** (Continued)

| Subgroups | PFOA | PFOS | PFHxS | PFDeA | PFNA | MPAH | ΣPFAS |
|---|---|---|---|---|---|---|---|
| p value | **0.007** | **0.01** | **0.001** | 0.4 | 0.2 | **0.019** | **0.005** |
| Drinking status | | | | | | | |
| Never | 3.50 (2.20,5.50) | 12.90 (7.20,20.60) | 1.50 (0.90,2.90) | 0.30 (0.14,0.40) | 1.00(0.70,1.70) | 0.30(0.12,0.50) | 20.58(12.57,31.32) |
| Former | 3.90 (2.50,5.60) | 14.40 (9.00,23.00) | 1.70 (1.00,3.00) | 0.30 (0.14,0.50) | 1.20(0.82,1.80) | 0.30(0.12,0.50) | 22.98(15.28,34.40) |
| Now | 4.10 (2.80,5.80) | 13.90 (8.70,22.00) | 1.90(1.10,3.20) | 0.30 (0.20,0.50) | 1.23(0.82,1.72) | 0.30 (0.12,0.50) | 23.36(15.30,33.64) |
| p value | **<0.01** | **0.05** | **0.02** | **0.02** | 0.06 | 0.84 | **0.03** |
| Year cycle | | | | | | | |
| 2005–2006 | 4.30 (2.80, 6.50) | 17.90 (11.80, 27.42) | 1.80 (1.00, 3.10) | 0.40 (0.20, 0.50) | 1.10 (0.77, 1.72) | 0.40 (0.30, 0.75) | 25.94 (16.68, 36.90) |
| 2007–2008 | 4.40 (3.10, 6.30) | 14.30 (9.20, 21.40) | 2.00 (1.10, 3.40) | 0.30 (0.14, 0.40) | 1.23 (0.90, 1.72) | 0.30 (0.12, 0.50) | 21.91 (14.30, 30.90) |
| 2009–2010 | 3.40 (2.30, 4.70) | 10.30 (6.45, 15.50) | 1.70 (1.00, 2.90) | 0.30 (0.20, 0.40) | 1.23 (0.90, 1.72) | 0.20 (0.10, 0.30) | 15.90 (10.21, 23.20) |
| p value | **<0.001** | **<0.001** | 0.091 | 0.055 | 0.3 | **<0.001** | **<0.001** |

Non-normally distributed continuous variables are presented as median (IQR)

PFAS [8, 33]. In addition, our results indicate that PFAS accumulation is more likely to occur in individuals with low levels of daily physical activity, poor diet quality, and a history of former or current smoking. This aligns with the existing evidence linking higher concentrations of PFAS to obesity [34] and is consistent with the patterns observed in our data. An intriguing finding from our study is the annual decrease in PFAS exposure among the US population between 2005 and 2010. This decline may be attributed to increased awareness of the toxicity associated with PFAS and subsequent regulatory actions taken to restrict their use [35].

Our research consistently confirmed the negative correlation between PFAS and constipation through logistic regression, subgroup analysis, and RCS results. Specifically, PFOA, PFOS, and PFHxS demonstrated a robust negative association with constipation. Previous studies have indicated that PFOA and PFOS have a high intestinal absorption index of over 90% in animal trials, while PFHxS can be almost completely absorbed from the intestine [8]. In our study, the negative correlation between PFOA and PFHxS with constipation persisted even after adjusting for multiple potential influencing factors in models A and B. However, this correlation weakened as more adjustment variables were included. This could be attributed to individual factors such as gender, age, race, diet, activity level, smoking, and alcohol consumption, which might influence the exposure and distribution of PFAS and thereby interfere with the results of univariate analysis. Nonetheless, we still observed a significant negative correlation in the fully adjusted model. Based on our findings, it can be presumed that PFAS may promote intestinal absorption and enhance peristalsis function, thereby increasing intestinal absorption and reducing the risk of constipation. The exact mechanism underlying this result remains unclear but it is thought to involve PFAS-induced changes in inflammatory factors and the gut microbiome (**Fig 5**).

To elucidate the negative association between PFAS and constipation, previous studies have indicated that exposure to PFOA can upregulate pro-inflammatory cytokines, such as TNF-α, IL-1β, and COX-2, while downregulating anti-inflammatory cytokines like IL-10 [36]. TNF-α, in particular, has been recognized as a key factor in inhibiting fluid absorption and promoting fluid secretion into the intestinal lumen [37, 38], ultimately increasing fecal moisture and volume. This mechanism may explain the observed correlation between PFOA and constipation. Moreover, the upregulation of IL-1β can induce increased intestinal tight-junction permeability, leading to inflammation [39]. Inflammation is known to predispose individuals to diarrhea, which could help explain the decreased incidence of constipation in relation to PFAS exposure.

**Table 3. Association between PFAS and risk of constipation.**

| PFAS | Unadjusted Model | | Model A | | Model B | |
|---|---|---|---|---|---|---|
| | OR(95% CI) | p value | OR(95% CI) | p value | OR(95% CI) | p value |
| **PFOA** | | | | | | |
| Continuous | 0.73(0.62,0.85) | **<0.001** | 0.78(0.64,0.96) | **0.021** | 0.80(0.67,0.97) | **0.021** |
| Q1 | ref | ref | ref | ref | ref | ref |
| Q2 | 0.63(0.48,0.84) | **0.002** | 0.76(0.50,1.18) | 0.209 | 0.72(0.51,1.02) | 0.064 |
| Q3 | 0.56(0.42,0.75) | **<0.001** | 0.68(0.48,0.95) | **0.025** | 0.67(0.49,0.91) | **0.013** |
| p for trend | | **<0.001** | | **0.023** | | **0.01** |
| **PFOS** | | | | | | |
| Continuous | 0.85(0.74,0.97) | **0.018** | 0.90(0.73,1.10) | 0.286 | 0.92(0.78,1.08) | 0.312 |
| Q1 | ref | ref | ref | ref | ref | ref |
| Q2 | 0.74(0.52,1.04) | 0.082 | 0.67(0.40,1.11) | 0.114 | 0.71(0.50,1.01) | 0.055 |
| Q3 | 0.65(0.48,0.88) | **0.006** | 0.74(0.49,1.11) | 0.143 | 0.77(0.54,1.10) | 0.141 |
| p for trend | | **0.006** | | 0.14 | | 0.137 |
| **PFHxS** | | | | | | |
| Continuous | 0.85(0.76,0.95) | **0.006** | 0.91(0.78,1.06) | 0.218 | 0.94(0.82,1.07) | 0.344 |
| Q1 | ref | ref | ref | ref | ref | ref |
| Q2 | 0.53(0.39,0.73) | **<0.001** | 0.58(0.38,0.87) | **0.011** | 0.58(0.41,0.81) | **0.002** |
| Q3 | 0.59(0.43,0.80) | **0.001** | 0.65(0.41,1.01) | 0.059 | 0.70(0.48,1.02) | 0.059 |
| p for trend | | **<0.001** | | 0.053 | | **0.049** |
| **PFDeA** | | | | | | |
| Continuous | 0.91(0.76,1.09) | 0.306 | 0.82(0.67,1.01) | 0.064 | 0.90(0.76,1.06) | 0.2 |
| Q1 | ref | ref | ref | ref | ref | ref |
| Q2 | 0.69(0.48,0.99) | **0.044** | 0.83(0.56,1.25) | 0.357 | 0.76(0.53,1.08) | 0.121 |
| Q3 | 0.84(0.62,1.12) | 0.224 | 0.77(0.56,1.05) | 0.092 | 0.86(0.65,1.13) | 0.259 |
| p for trend | | 0.149 | | 0.067 | | 0.189 |
| **MPAH** | | | | | | |
| Continuous | 0.99(0.85,1.14) | 0.865 | 1.02(0.85,1.24) | 0.801 | 0.98(0.82,1.17) | 0.78 |
| Q1 | ref | ref | ref | ref | ref | ref |
| Q2 | 0.97(0.75,1.24) | 0.779 | 1.19(0.83,1.71) | 0.34 | 1.03(0.78,1.35) | 0.857 |
| Q3 | 1.14(0.81,1.62) | 0.441 | 1.41(0.89,2.23) | 0.142 | 1.18(0.77,1.80) | 0.444 |
| p for trend | | 0.472 | | 0.14 | | 0.454 |
| **PFNA** | | | | | | |
| Continuous | 0.92(0.74,1.16) | 0.486 | 0.93(0.74,1.17) | 0.535 | 0.97(0.79,1.19) | 0.769 |
| Q1 | ref | ref | ref | ref | ref | ref |
| Q2 | 0.92(0.63,1.33) | 0.635 | 1.11(0.73,1.70) | 0.617 | 1.08(0.74,1.57) | 0.68 |
| Q3 | 0.79(0.53,1.16) | 0.221 | 0.78(0.53,1.15) | 0.198 | 0.88(0.61,1.28) | 0.502 |
| p for trend | | 0.226 | | 0.227 | | 0.527 |
| **ΣPFAS** | | | | | | |
| Continuous | 0.78(0.67,0.92) | **0.003** | 0.84(0.67,1.06) | 0.135 | 0.88(0.73,1.06) | 0.178 |
| Q1 | ref | ref | ref | ref | ref | ref |
| Q2 | 0.75(0.55,1.03) | 0.07 | 0.75(0.50,1.11) | 0.144 | 0.76(0.56,1.04) | 0.079 |
| Q3 | 0.58(0.45,0.77) | **<0.001** | 0.69(0.48,1.01) | 0.059 | 0.70(0.52,0.95) | **0.022** |
| p for trend | | **<0.001** | | 0.055 | | **0.019** |

Continuous variables are logarithmic transformed.

Categorical variables are weighted quartiles of the raw data.

Unadjusted Model: did not adjust for any potential confounding factors.

Model A: was adjusted for age, gender, race, body mass index, educational level, marital status, PIR.

Model B, was adjusted for age, gender, race, body mass index, educational level, marital status, PIR, diet pattern, activity, smoke status and alcohol consumption.

ORs (95% CI): The odds ratios and corresponding 95% confidence intervals.

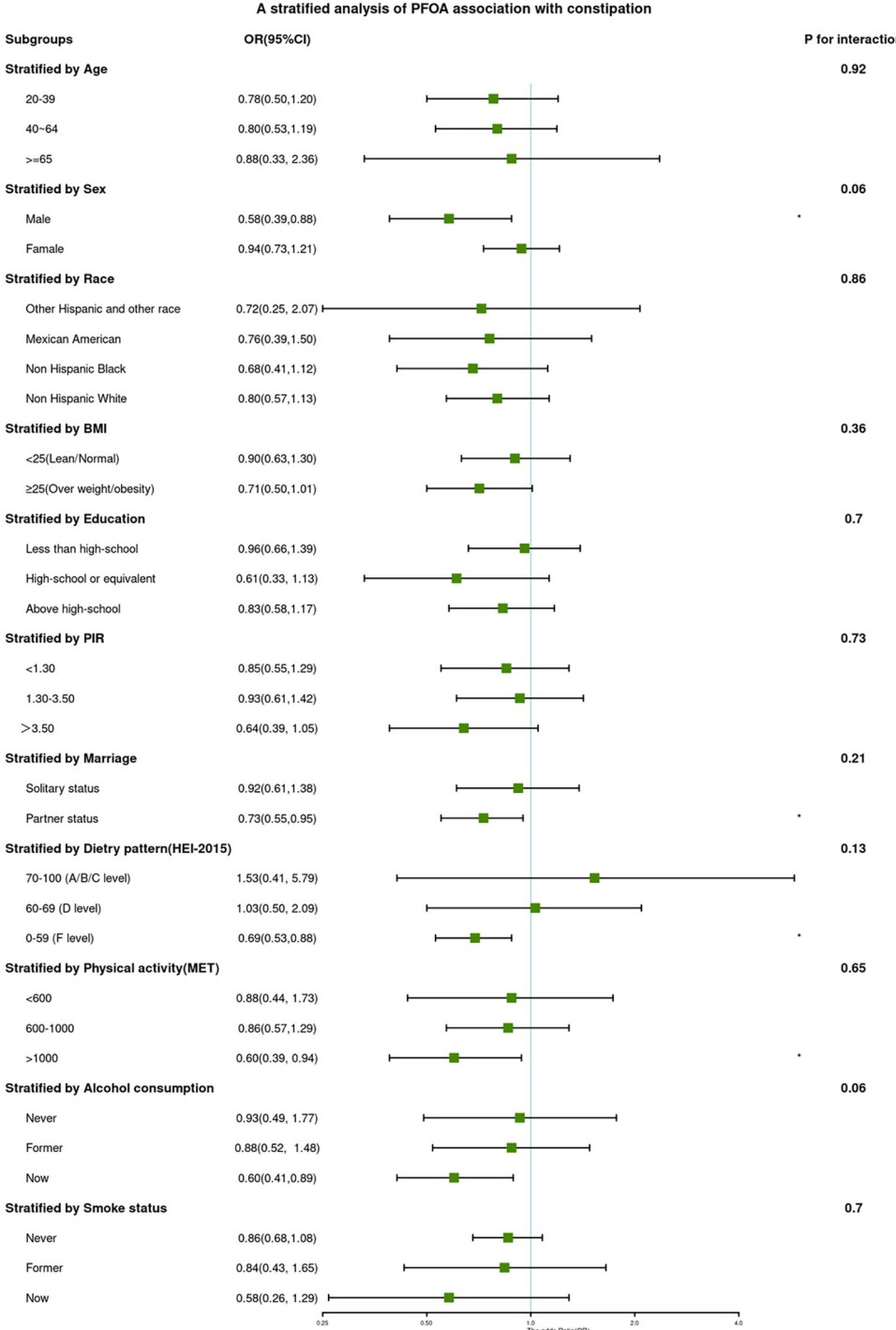

**Fig 2. ORs (95% CI) from multiple logistic regression analysis models of associations between PFOA and risk of constipation in different subgroups.**

An alternate explanation for the observed results may involve the mediation of gut microbial changes by PFAS exposure [40]. Multiple studies have suggested this possibility [36, 41–43]. Notably, PFAS exposure has been reported to decrease the abundance of *Dehalobacterium* and *Bacteroides* [42], whereas a significant decrease in the *firmicute*/*Bacteroides* ratio is

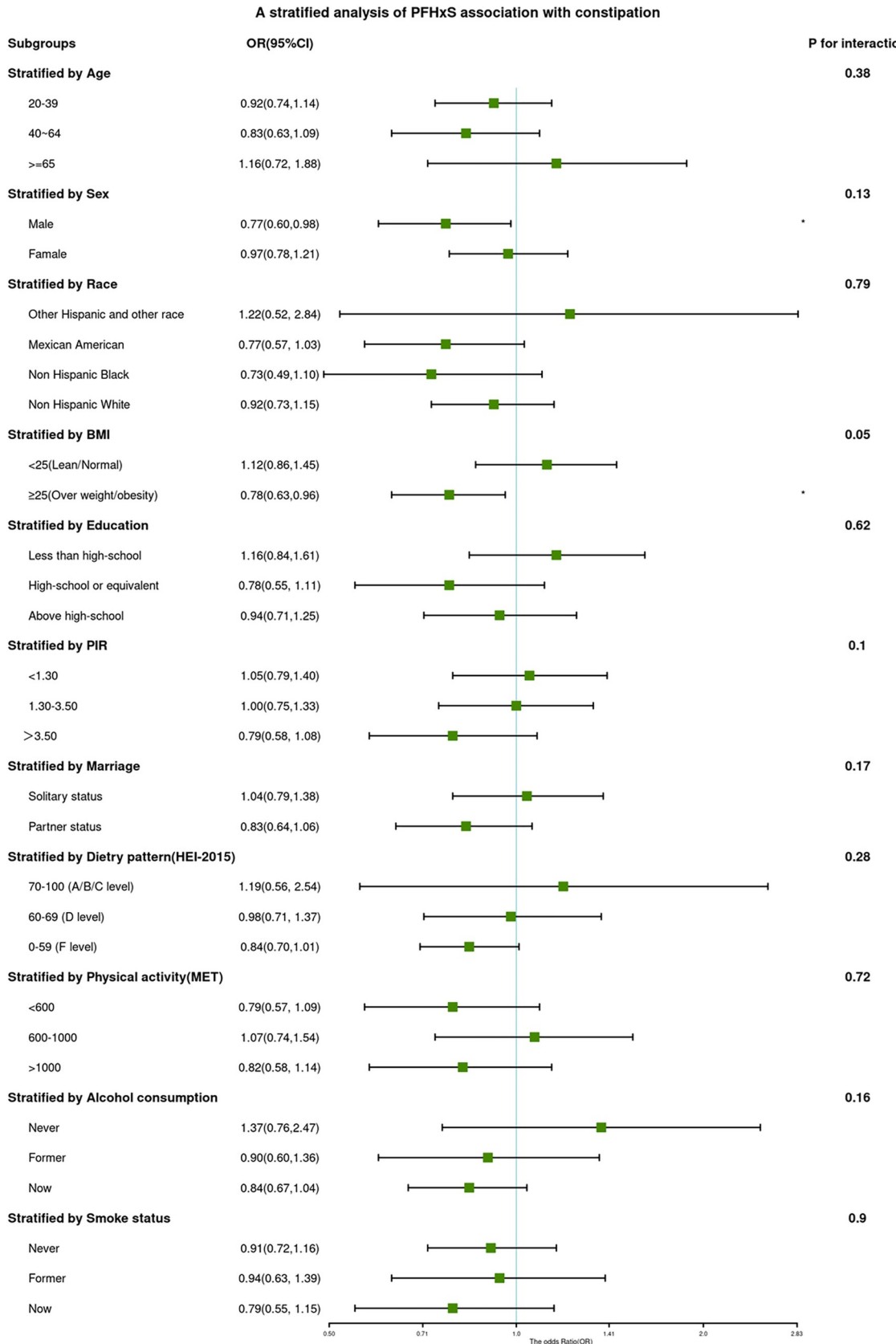

A stratified analysis of PFHxS association with constipation

| Subgroups | OR(95%CI) | | P for interaction |
|---|---|---|---|
| **Stratified by Age** | | | 0.38 |
| 20-39 | 0.92(0.74,1.14) | | |
| 40~64 | 0.83(0.63,1.09) | | |
| >=65 | 1.16(0.72, 1.88) | | |
| **Stratified by Sex** | | | 0.13 |
| Male | 0.77(0.60,0.98) | | * |
| Famale | 0.97(0.78,1.21) | | |
| **Stratified by Race** | | | 0.79 |
| Other Hispanic and other race | 1.22(0.52, 2.84) | | |
| Mexican American | 0.77(0.57, 1.03) | | |
| Non Hispanic Black | 0.73(0.49,1.10) | | |
| Non Hispanic White | 0.92(0.73,1.15) | | |
| **Stratified by BMI** | | | 0.05 |
| <25(Lean/Normal) | 1.12(0.86,1.45) | | |
| ≥25(Over weight/obesity) | 0.78(0.63,0.96) | | * |
| **Stratified by Education** | | | 0.62 |
| Less than high-school | 1.16(0.84,1.61) | | |
| High-school or equivalent | 0.78(0.55, 1.11) | | |
| Above high-school | 0.94(0.71,1.25) | | |
| **Stratified by PIR** | | | 0.1 |
| <1.30 | 1.05(0.79,1.40) | | |
| 1.30-3.50 | 1.00(0.75,1.33) | | |
| >3.50 | 0.79(0.58, 1.08) | | |
| **Stratified by Marriage** | | | 0.17 |
| Solitary status | 1.04(0.79,1.38) | | |
| Partner status | 0.83(0.64,1.06) | | |
| **Stratified by Dietry pattern(HEI-2015)** | | | 0.28 |
| 70-100 (A/B/C level) | 1.19(0.56, 2.54) | | |
| 60-69 (D level) | 0.98(0.71, 1.37) | | |
| 0-59 (F level) | 0.84(0.70,1.01) | | |
| **Stratified by Physical activity(MET)** | | | 0.72 |
| <600 | 0.79(0.57, 1.09) | | |
| 600-1000 | 1.07(0.74,1.54) | | |
| >1000 | 0.82(0.58, 1.14) | | |
| **Stratified by Alcohol consumption** | | | 0.16 |
| Never | 1.37(0.76,2.47) | | |
| Former | 0.90(0.60,1.36) | | |
| Now | 0.84(0.67,1.04) | | |
| **Stratified by Smoke status** | | | 0.9 |
| Never | 0.91(0.72,1.16) | | |
| Former | 0.94(0.63, 1.39) | | |
| Now | 0.79(0.55, 1.15) | | |

The odds Ratio(OR)

**Fig 3. ORs (95% CI) from multiple logistic regression analysis models of associations between PFHxS and risk of constipation in different subgroups.**

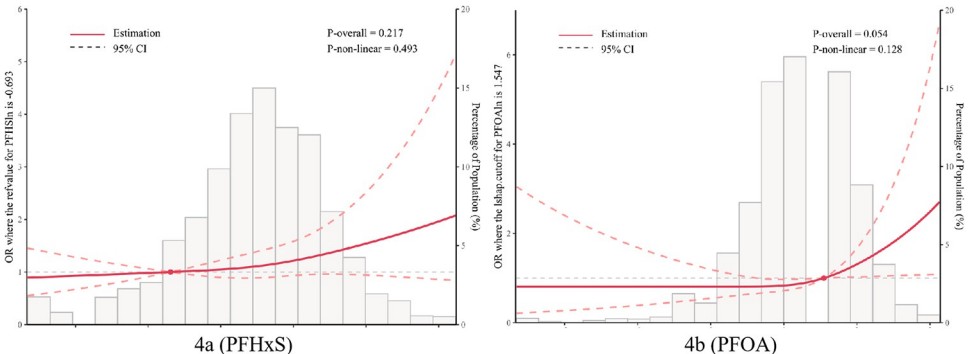

**Fig 4.** Restricted cubic spline (RCS) models for the relationship between PFHxS (a), PFOA (b) and the risk of constipation.

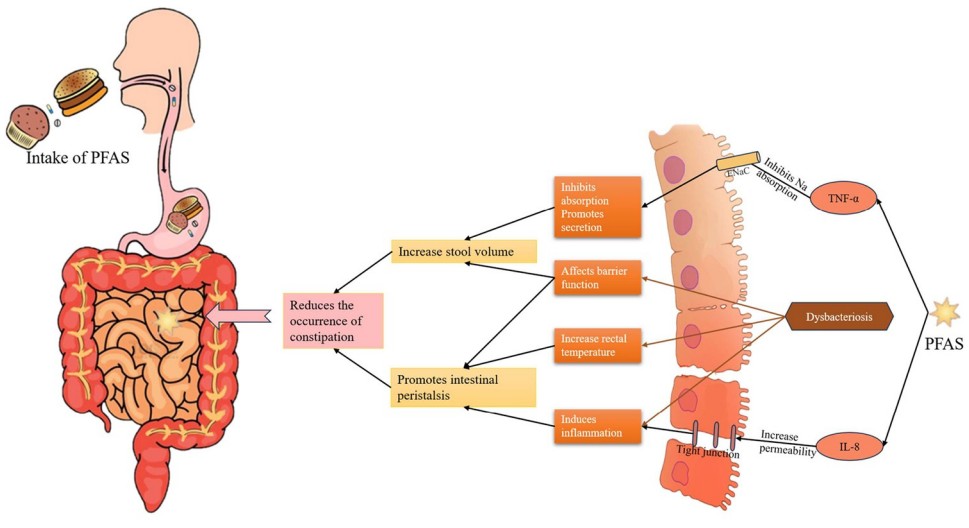

*ENaC: The epithelial Na channel

**Fig 5. Diagram of the mechanism by which PFAS causes a decrease in constipation.**

considered a key marker of intestinal dysbiosis [8]. Thus, PFAS exposure may potentially increase the *Firmicute/Bacteroides* ratio, serving as a protective factor in regulating constipation. Furthermore, a high-fat diet, which effectively elevates PFOA levels in the body [44], has been associated with a reduced abundance of *Catenibacterium mitsuokai [40] Catenibacterium mitsuokai* has been found to negatively correlate with rectal temperature [45], suggesting that PFOA could potentially increase rectal temperature by disrupting the abundance of *Catenibacterium mitsuokai* and improving anorectal motility. However, it is important to note that our findings contrast with those of Kelsey N. Thompson et al., who found that PFOS exposure increases the abundance of pro-inflammatory flora such as *Bilophila wadsworthia* and *Odoribacter splanchnicus*, while promoting the growth of *Faecalibacterium prautzii*, which has been significantly associated with constipation-predominant irritable bowel syndrome (IBS-C) [46]

Research has found that PFAS exposure is positively associated with the occurrence and development of various diseases. The mechanisms through which PFAS harm different systems vary. It is believed that PFAS exposure causes immune suppression, particularly in relation to reduced vaccine response and increased risk of certain infectious diseases or symptoms [47]. PFAS exposure is also associated with increased risk of cardiovascular diseases [48], This may be due to the effects of PFAS on blood lipid levels [49], blood pressure [50], and atherosclerosis [51], leading to abnormal functioning of the cardiovascular system. Studies have proposed an association between PFAS exposure and fatty degeneration [52], as well as a significant positive correlation with liver fibrosis [53]. The mechanisms behind these associations may involve disruption of liver cell metabolism and oxidative stress processes [54, 55]. In more severe cases, PFAS exposure is associated with increased risk of certain cancers such as kidney cancer, testicular cancer, prostate cancer, and breast cancer, this is generally believed to be related to oxidative stress, the mutagenic properties of PFAS, and their hormone-disrupting effects [56, 57]. Conversely, some diseases have been found to be negatively associated with PFAS exposure, such as cardiovascular diseases [49], and chronic kidney disease [58], researchers suggest that this may be due to the high oxygen-carrying capacity of PFAS. In our study, we observed a negative correlation between PFAS exposure and constipation. Based on the studies we identified, we believe this may be related to imbalanced gut microbiota, increased intestinal motility, liver metabolism abnormalities caused by oxidative stress, and inflammatory responses. For future research, we recommend implementing cohort studies to establish the connection between PFAS exposure and constipation and diarrhea. Furthermore, the underlying mechanism linking constipation with PFAS remains elusive, warranting further investigation into intestinal flora and inflammatory factors. Such studies could provide valuable insights into intestinal health and shed light on the role of environmental factors in influencing intestinal symptoms.

Our has several notable strengths. Firstly, it utilized the NHANES database, which provides a comprehensive range of data on both environmental exposure and basic participant information. The standardized study protocol and robust quality control measures ensure the reliability and accuracy of the data. Secondly, our study also contributes by examining the distribution of PFAS compounds, which provides a comprehensive assessment of their accumulation and distribution patterns. This valuable information improves our understanding of these compounds and aids in guiding personalized protective measures. Furthermore, we employed three different adjustment models to investigate the association between PFAS and constipation. This approach allows for a more comprehensive interpretation of the results by considering various potential confounding factors. Lastly, the use of restrictive cubic spline (RCS) analysis enabled us to explore the potential nonlinear relationship between PFAS and constipation. This approach provides a more comprehensive understanding of the dose-response relationship and allows for a more nuanced interpretation of the findings. This study has several limitations that need to be considered. Firstly, its cross-sectional design restricts our ability to establish causal relationships or investigate temporal trends between PFAS and constipation. Instead, it enables us to examine cumulative effects. Secondly, the diagnoses relied on self-reported information, which may have introduced recall bias.

## 5. Conclusion

The key finding of this study is the inverse correlation observed between serum PFAS, particularly PFOA and PFHxS, and constipation. The underlying mechanism behind this association may involve changes in intestinal flora, inflammatory factors, and cytokines. However, due to the cross-sectional nature of our study, it is imperative that further cohort studies and basic

research be conducted to elucidate the relationship between PFAS and constipation, its underlying mechanisms, and explore any potential associations with diarrhea.

## Acknowledgments

The authors thank the National Center for Health Statistics for planning and directing the NHANES.

## Author Contributions

**Conceptualization:** Yifan Zhao, Ke Pu, Ya Zheng, Yuping Wang, Jun Wang, Yongning Zhou.

**Formal analysis:** Yifan Zhao, Ya Zheng, Jun Wang.

**Funding acquisition:** Ya Zheng, Yongning Zhou.

**Investigation:** Yifan Zhao, Ya Zheng, Jun Wang.

**Methodology:** Ke Pu, Ya Zheng, Jun Wang.

**Supervision:** Jun Wang, Yongning Zhou.

**Writing – original draft:** Yifan Zhao, Ya Zheng, Yuping Wang.

**Writing – review & editing:** Ke Pu, Jun Wang, Yongning Zhou.

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
