## [Decision Letter · Decision Letter 0]

29 Jan 2024

PONE-D-23-31213Association between PFAS and constipation: Results from NHANES 2005-2010PLOS ONE

Dear Dr. Zhou,

Thank you for submitting your manuscript to PLOS ONE. After careful consideration, we feel that it has merit but does not fully meet PLOS ONE’s publication criteria as it currently stands. Therefore, we invite you to submit a revised version of the manuscript that addresses the points raised during the review process.

We look forward to receiving your revised manuscript.

Kind regards,

Misbahuddin Rafeeq

Academic Editor

PLOS ONE

Journal Requirements:

Additional Editor Comments:

The authors have addressed an important issue regarding the possible association of PFAS with constipation. The study design, methodology and manuscript draft seems fair enough. However, few minor points need to be addressed. Additionally, a limitation section can be added at the end of the manuscript before the conclusion highlighting the key shortcomings which can be improved in further studies on the subject.

Reviewers' comments:

Reviewer's Responses to Questions

**Comments to the Author**

1. Is the manuscript technically sound, and do the data support the conclusions?

Reviewer #1: Yes

2. Has the statistical analysis been performed appropriately and rigorously? 

Reviewer #1: I Don't Know

3. Have the authors made all data underlying the findings in their manuscript fully available?

Reviewer #1: Yes

4. Is the manuscript presented in an intelligible fashion and written in standard English?

Reviewer #1: Yes

5. Review Comments to the Author

Reviewer #1: This article explores an interesting issue of ‘Association between PFAS and constipation’, and tries to fill the knowledge gap. I acknowledge and appreciate the efforts of the authors and research team. My propositions are as below:

Minor comments:

• It is suggested not to use only abbreviations in the title such as PFAS and NHANES. Years 2005-2010 can also be out in brackets.

• The study data timeframe (2005-2010) is mentioned. It can also be informative to share when specifically, it was extracted for this study.

• The adverse effects of PFAS on various health issues, including immunological, metabolic, cardiovascular, liver, renal problems, and cancer are mentioned in the introduction section. However, the issue needs to be elaborated in the discussion section to explore and consider the adverse and beneficial roles of PFAS.

• For better readers’ understanding, the OR needs to be mentioned in the relevant column titles such as in Table 3 (as mentioned in table captions).

• In line 137, it is not clear whether the defined group ABC or groups A, B, and C represent a better level of diet quality.

• A similar and consistent logical sequence of major findings in the results and discussion section can be considered.

6. PLOS authors have the option to publish the peer review history of their article (what does this mean?). If published, this will include your full peer review and any attached files.

Reviewer #1: No

---

## [Author Response · Author response to Decision Letter 0]

6 Mar 2024

Dear editor and reviewers,

Thank you for giving us the opportunity to revise our manuscript. We have addressed your comments and thoroughly revised the manuscript by providing a point-by-point response as shown below.

We hope the revised version satisfy the publishing requirements of Plos One.

We look forward to your final decision on this manuscript.

Sincerely,

Yongning Zhou

A point-by-point response to the comments

Journal Requirements:

Response: Thank you for bringing this to my attention. I have carefully reviewed PLOS ONE's style and file naming requirements and made the necessary adjustments to ensure the manuscript complies with all guidelines. I appreciate your patience and understanding.

2. Please ensure that you have an ORCID iD and that it is validated in Editorial Manager.

Response: Thank you for your reminder. Corresponding author’s ORCID has been successfully authorized and validated in the Editorial Manager. 

Response: Apologize for the mistake. We have already provided funding information in "Funding information" part in submission system and verified its accuracy. And we have provided detailed funding disclosures in the cover letter.

4. Your ethics statement should only appear in the Methods section of your manuscript.

Response: Thank you for your guidance. We have moved the ethics statement to the Methods. This can be found on lines 104-107, page 4. 

5. Please review your reference list to ensure that it is complete and correct.

Response: Thank you. We have thoroughly reviewed our reference list to ensure its completeness and accuracy. 

Additional Editor Comments:

The authors have addressed an important issue regarding the possible association of PFAS with constipation. The study design, methodology and manuscript draft seems fair enough. However, few minor points need to be addressed. Additionally, a limitation section can be added at the end of the manuscript before the conclusion highlighting the key shortcomings which can be improved in further studies on the subject.

Response: Thank you for your constructive feedback. Limitations have been added before the conclusion. This can be found on lines 386-390, page 20. 

Comments from Reviewer 1

This article explores an interesting issue of ‘Association between PFAS and constipation’, and tries to fill the knowledge gap. I acknowledge and appreciate the efforts of the authors and research team. My propositions are as below: 

Minor comments:

1. It is suggested not to use only abbreviations in the title such as PFAS and NHANES. Years 2005-2010 can also be out in brackets.

Response: Thank you for your valuable suggestions. Modifications have been made in accordance with your suggestions to create a more clear title.

2. The study data timeframe (2005-2010) is mentioned. It can also be informative to share when specifically, it was extracted for this study. 

Response: Thank you for pointing out this. We have specified the precise data extraction time in the first part of the “Method” part. This can be found in line 91, page 4. 

3. The adverse effects of PFAS on various health issues, including immunological, metabolic, cardiovascular, liver, renal problems, and cancer are mentioned in the introduction section. However, the issue needs to be elaborated in the discussion section to explore and consider the adverse and beneficial roles of PFAS. 

Response: Apologize for not having fully elaborated earlier. Thank you for pointing out this. We have further discussed our findings in light of the impact and underlying mechanisms of PFAS on the immune system, metabolism, cardiovascular health, liver and kidney issues, and cancer, respectively. This can be found on line 350-368, page 19. 

4. For better readers’ understanding, the OR needs to be mentioned in the relevant column titles such as in Table 3 (as mentioned in table captions). 

Response: Apologize for this careless mistake. We have made necessary changes to the table 3 as per your recommendation.

5. In line 137, it is not clear whether the defined group ABC or groups A, B, and C represent a better level of diet quality. 

Response: Apology for confusing. Existing research does not show a significant difference in dietary quality between Group ABC and the other two groups. Thus, we have made corrections to the erroneous portions of the text. This can be found on lines 150-152, page 6. 

6. A similar and consistent logical sequence of major findings in the results and discussion section can be considered. 

Response: Thank you for your constructive suggestions. The title of the section on results: "3.2 Distribution characteristics of PFAS in different subgroups," may have caused some confusion. We have revised it to "3.2 Distribution characteristics of PFAS" to ensure that the narrative in the results section is consistent with that in the discussion section. 

To further align the structure of the discussion section to make it a similar and consistent logical sequence with the major findings, we have reorganized the first paragraph as a summary of the main findings, the second paragraph discusses population characteristics, the third paragraph delves into the discussion of "Distribution characteristics of PFAS," and we have combined the results from logistic regression, subgroup analysis, and RCS into paragraphs 4, 5, and 6 for a more comprehensive discussion. Paragraph 7 discusses the associations between PFAS and other systemic diseases, as well as future prospects. Finally, paragraphs 8 and 9 are dedicated to discussing the strengths and limitations of this study.

---

## [Editor Report · Decision Letter 1]

12 Mar 2024

Association of per- and polyfluoroalkyl substances with constipation: The National Health and Nutrition Examination Survey (2005-2010)

PONE-D-23-31213R1

Dear Dr. Yongning Zhou,

We’re pleased to inform you that your manuscript has been judged scientifically suitable for publication and will be formally accepted for publication once it meets all outstanding technical requirements.

Kind regards,

Misbahuddin Rafeeq

Academic Editor

PLOS ONE
---

## [Editor Report · Acceptance letter]

22 Mar 2024

PONE-D-23-31213R1 

PLOS ONE

Dear Dr. Zhou, 

I'm pleased to inform you that your manuscript has been deemed suitable for publication in PLOS ONE. Congratulations! Your manuscript is now being handed over to our production team.

Kind regards, 

on behalf of

Dr. Misbahuddin Rafeeq 

Academic Editor

PLOS ONE